# Fabrication of Selective Thermal Emitter with Multilayer Films for Mid-/Low-Temperature Infrared Stealth with Radiative Cooling

**Mengdan Qian [1], Qingqing Shi [1], Lin Qin [1], Jinlong Huang [1], Caixia Guo [2], Yufang Liu [1] and Kun Yu [1,***

[1] Henan Key Laboratory of Infrared Materials & Spectrum Measures and Applications, School of Physics, Henan Normal University, Xinxiang 453007, China; qianmengdan@htu.edu.cn (M.Q.)
[2] College of Electronic and Electric Engineering, Henan Normal University, Xinxiang 453007, China
[*] Correspondence: yukun@htu.edu.cn

**Abstract:** Infrared selective emitters are attracting more and more attention due to their modulation ability of infrared radiance, which provides an efficient ability to blend objects into the surrounding environment. In this paper, an Ag/ZnS/Si/Ag/Si multilayered emitter is proposed by virtue of impedance matching as well as Fabry-Perot cavity effect to achieve selective radiation in the infrared band. The emissivity of the fabricated selective emitter is measured to be $\varepsilon_{3-5\mu m} = 0.16$ and $\varepsilon_{8-14\mu m} = 0.23$ in the atmosphere windows, respectively, meeting the requirements of infrared stealth. Meanwhile, the emissivity at the non-atmospheric window (5–8 μm) is as high as 0.78, which allows efficient heat dissipation to achieve radiative cooling. Furthermore, the selective emitter maintains excellent stealth performance until 350 °C, indicating its good heat resistance and dissipation at medium temperature. The proposed emitter with spectral selectivity provides a new strategy for the facile fabrication of mid-/low-temperature infrared stealth devices.

**Keywords:** multilayer films; selective emitter; infrared stealth; thermal dissipation; radiative cooling





## 1. Introduction

Modulation of radiation characteristics facilitates the development of selective thermal emitters that have great potential applications in fields of infrared stealth, radiative cooling and energy utilization [1–5]. Infrared (IR) stealth technology is widely utilized in modern military confrontation due to its capacity for signal evasion and survival from infrared detectors and imaging systems [6]. Considering that all objects radiatively emit thermal emission signals, they are more likely to be exposed to infrared detection configurations. Thus, the infrared stealth technology is devoted to reducing the IR radiation difference between targets and the surrounding environment, making them "invisible" to the detector [7,8]. According to the Stephen-Boltzmann law $P = \varepsilon\sigma T^4$, where σ is the Steven-Boltzmann constant, $\varepsilon$ and T refer to the surface emissivity and the absolute temperature of the tested object. It is clear that the thermal radiation energy (P) is proportional to the surface emissivity and the fourth power of its absolute temperature. Compared to the complicated temperature-control circuits and equipment to change the local temperature, selective emissivity engineering is a more convenient way to regulate the thermal radiation energy [9–11].

Military IR detection devices usually work during the infrared band of 3–5 μm and 8–14 μm, corresponding to the atmospheric windows with high transmittance for electromagnetic waves [12–14]. In order to achieve thermal camouflage, the surface emissivity of objects should be tuned to low values in the atmospheric windows to blend into the environment. However, low emissivity in the whole range of the IR regime may lead to a rise in surface temperature and then thermal energy accumulation, which is unfavorable for stealth performance. To solve this problem, a selective thermal radiation scheme is

demonstrated for infrared stealth [15]. The selective emitter requires low emissivity in the atmospheric band (3–5 μm and 8–14 μm) to evade detection while increasing thermal radiance in the undetected band (5–8 μm) for energy dissipation [16–18]. Based on this principle, various selective emitters with functional micro/nanostructures, such as multi-layer films, photonics crystals, gratings and metamaterials, are demonstrated for infrared camouflage applications [19–21]. For instance, Li et al. put forward a wavelength selective emitter based on Ge/ZnS multilayer films, which shows good IR camouflage performance and efficient thermal management at high temperature [22]. Cho et al. demonstrate a meta-material selective emitter with infrared camouflage and energy dissipation properties [23]. In comparison to existing stealth devices with specific micro/nanostructures that require complicated processing techniques, multilayer-film based emitters are more practically realizable due to the virtue of simple structures and low costs [19,24]. However, it remains a big challenge to optimize the selective emitter with more simplified film structure as well as improved camouflage performance.

In this work, a five-layer film based selective thermal emitter (Ag/ZnS/Si/Ag/Si) is proposed for infrared stealth by virtue of the radiation characteristic of ultrathin Ag layer and impedance matching. The emitter shows advantages of structural simplicity and superior spectral selectivity during the 3–14 μm band. According to the results of measured spectral emissivity, the selective thermal emitter has suppressed radiation at two atmospheric windows (3–5 μm and 8–14 μm) for the requirement of infrared stealth. Meanwhile, the emissivity of selective emitter at non-atmospheric band (5–8 μm) is as high as 0.78, indicating the good thermal dissipation capacity. The emitter maintains stable working performance when the ambient temperature is below 400 °C, which is attributed to its heat dissipation and radiative cooling ability. This work provides a facile strategy for the fabrication of highly scalable and practical selective emitters for low-/mid-temperature infrared stealth technology.

## 2. Materials and Methods

Numerical simulations: the optical simulation was performed with a wave optical module (electromagnetic waves, frequency domain) in COMSOL Multiphysics 6.0 software. The governing equation was Maxwell's equation and it was discretized for a finite element method. The simulated domain consisted of a metal ground (Ag), a dielectric layer (ZnS), a lower dielectric layer (Si), a metal layer (Ag), an upper dielectric layer (Si) and a medium (Air). The parameters of the structure were defined in the text. We adopted the Lorentz-Drude model for analyzing the electromagnetic behaviors of Ag. The refractive index in COMSOL Multiphysics 6.0 software of ZnS and Si were used to model for dielectric layers. The real and imaginary parts of the refractive index of medium were assumed to be 1 and 0, respectively. Planar electromagnetic waves at the two ports were assumed, using periodic boundary conditions for the side walls. The emissivity (E) of the multilayer films is determined by calculating the absorptivity (A) according to the Kirchhoff's law which states that the spectral absorptivity and emissivity must be equal to each other (E = A).

The sample fabrication: the multilayer emitter and reference sample were prepared by electron beam evaporation technique on silicon substrates (1 inch). The deposition rate and film thickness were monitored by a quartz crystal oscillator. The target five-layer emitter was finally deposited under the evaporation rate of 0.02 nm/s for silver, 0.05–0.07 nm/s for zinc sulfide, and 0.05 nm/s for silicon when the vacuum in the chamber was pumped to $4.0 \times 10^{-4}$ Pa.

Infrared Spectral Emissivity Measurement: the spectral emissivity of the selective emit-ter at room temperature was measured by the reflectance accessory of Fourier-transform infrared spectrometer (FTIR). The measurement of spectral emissivity at different tempera-tures was performed by the mid-temperature spectral emissivity measuring device built in the laboratory. The blackbody (IR-563) is used as the standard radiation source. The thermal radiation was collected in the beam path of the FTIR spectrometer (VERTEX 70 V,

BRUKER, Germany) and detected by the liquid nitrogen-cooled mercury telluride (MCT) detector (BRUKER, Germany) with a spectral detection region ranging from 2 μm to 25 μm.

Infrared Thermal Image Measurement: the infrared thermal images were obtained by the thermal imaging camera (FLUKE TIX640, USA) working in the band of 7.5–14 μm over a range of temperatures in the practical environment.

## 3. Results and Discussion

### 3.1. Structural Design and Mechanism

Based on impedance matching theory, a thermal emitter with selectively high emissivity in 5–8 μm is proposed by integrating reflective silver layer with high refractive index materials of silicon (Si) and zinc sulfide (ZnS). The geometric structure and parameters of each layer are particularly explored to achieve the optimal infrared stealth performance. The schematic of designed selective emitter used for infrared camouflage is shown in Figure 1a. As shown, the emitter consists of five-layer planar films with an Ag film as the bottom reflective layer. The Si and ZnS layers are used as dielectric spacers sandwiched between the bottom and another ultrathin Ag intermediate layer, on which a dielectric Si layer is coated as the top layer. The selective emitter is finally designed as an Ag/ZnS/Si/Ag/Si five-layer structure where the thickness bottom Ag layer is set as 100 nm. In addition, the middle dielectric spacers of ZnS and Si layer have the same thickness of 410 nm, and the top Ag and Si layer are determined to be 7 nm and 350 nm, respectively. The selective emitter is experimentally fabricated based on the above optimal parameters using electron beam evaporation technology. As shown in the dotted box of Figure 1a, the cross-section view of fabricated emitter is characterized by scanning electron microscopy, and it can be observed that the thickness of each layer coincides well with the designed parameters. In order to meet the requirement of infrared camouflage, the spectral emissivity is expected to have an ideal value close to zero in the detected band (3–5 μm and 8–14 μm) while it keeps to the maximum throughout the entire 5–8 μm (dash line in Figure 1b). The calculated and measured spectral emissivity of the designed selective emitter are plotted in Figure 1b, which show similar spectral trend with the ideal one. The emissivity of fabricated emitter is measured to be $\varepsilon_{3–5\mu m} = 0.16$ and $\varepsilon_{8–14\mu m} = 0.23$ in the in the two atmospheric windows and it reaches 0.78 in the undetected band. The experimental emissivity corresponds well with the simulated result, demonstrating the reasonable structure design and potential stealth performance.

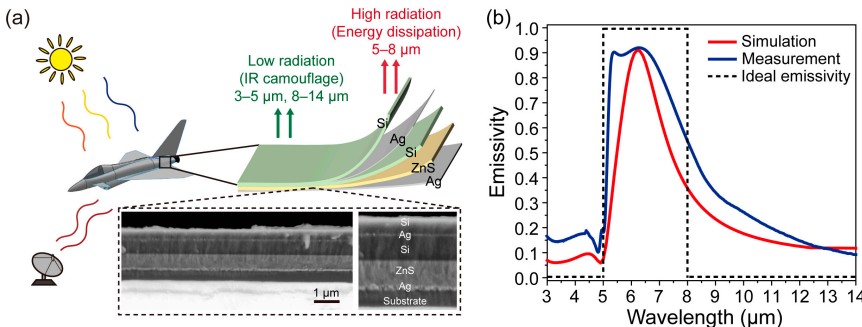

**Figure 1.** (**a**) Schematics of infrared detection process and infrared stealth selective emitter (from bottom to top is: Ag/ZnS/Si/Ag/Si); (**b**) Ideal selective emission (black dash line) for infrared stealth, simulated emissivity (blue line) and measured emissivity (red line) of the selective emitter.

Considering that the absorption behavior of selective emitters is susceptible to the geometric configuration, numerical analysis is subsequently performed to further investigate the effects of geometrical parameters on spectral behaviors. The Si wafer is used as the substrate in the experiment, and its thickness is much larger than the collective depth of the skin, which is capable of blocking the light passing through the whole structure. The bottom Ag film functions as a reflecting layer for infrared light, which has little influence on

the radiative characteristics of the emitter. Therefore, the thickness of the bottom reflected Ag layer is coated with the thickness of 100 nm. The influence of the other four layers on the absorption spectra is studied under different geometric parameters, and the simulated results are shown in Figure 2 and Figure S1. Specifically, the resonant wavelength redshifts slightly in 5–8 μm as the thickness of intermediate ZnS layer increases from 310 nm to 510 nm (Figure 2a). The thickness of the ZnS layer is finally determined to be 410 nm because deviation of this value will lead to slow increase of emissivity at 3–5 μm, which is undesirable for infrared stealth. Similarly, with the increase in the middle Si layer, the resonant wavelength redshifts clearly and the resonance intensity becomes stronger (Figure 2b). The resonance intensity becomes weaker as the film thickness reduces below 410 nm, and the selective emitter shows a narrowing absorption bandwidth within 5–8 μm when increasing the film thickness. Varying the thickness of top Ag layer from 5 nm to 21 nm, as observed in Figure 2c, results in the blueshift and narrowing of resonant absorption. Thus, the intermediate Ag layer thickness is determined to be 7 nm to achieve strong and broad resonant absorption in the 5–8 μm band. Furthermore, the absorption bandwidth increases continuously with the increase in the top Si thickness from 250 nm to 450 nm. However, the emittance of the emitter at 8–14 μm is high as the Si layer exceeds 350 nm, which is not favorable to infrared stealth. Taking all factors into consideration, the thickness of the top Si layer is set as 350 nm so that its absorption reaches the maximum in the non-atmospheric window while the radiation is suppressed effectively in the atmospheric window.

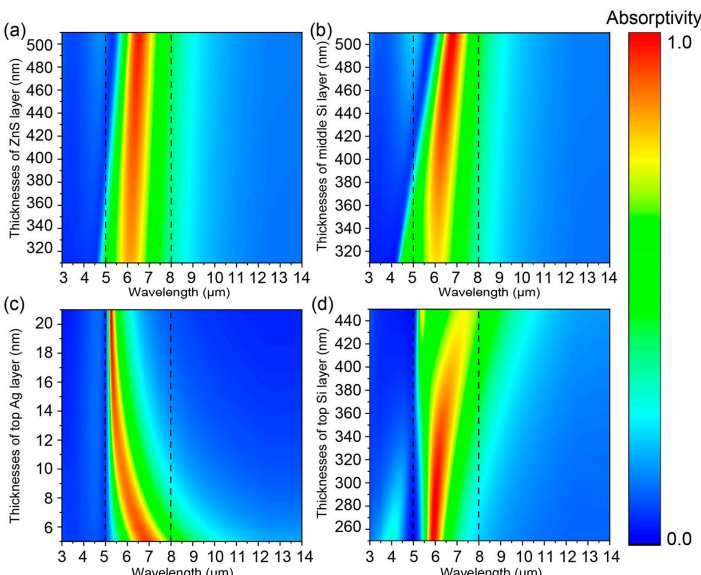

**Figure 2.** Dependence of the simulated absorption spectra on different geometric parameters with the thickness of (**a**) ZnS layer, (**b**) middle Si layer, (**c**) top Ag layer and (**d**) top Si layer of the selective emitter.

To demonstrate the simplicity, necessity and working performance of the designed five-layer selective emitter, the spectral behavior of the emitter configuration with different numbers of film layers is also investigated for comparison. Figure 3a gives the absorption spectra of the thermal emitter with Ag/ZnS/Si and Ag/ZnS/Si/Ag resonators. It is clear that the simple three-layer Ag/ZnS/Si emitters show little spectral selectivity over the whole infrared region. The introduction of ultrathin Ag film makes a Fabry-Perot (FP) cavity in the bottom Ag layer, resulting in the typical resonant absorption of Ag/ZnS/Si/Ag emitter at 5–8 μm. The position and intensity of resonant peak can be influenced by the thickness of the ultrathin Ag layer. Even though introduction of a 7 nm Ag layer results in a strong resonant peak near 5.6 μm, the resonant absorption is too narrow to achieve desired strong thermal emission in the undetected 5–8 μm band. The reduced film layers allow simplified emitter configuration but poor radiative performance for infrared stealth. Therefore, the Si film is introduced as the top layer, finally forming the Ag/ZnS/Si/Ag/Si

five-layer resonator. The top Si layer plays a crucial role in the bandwidth regulation because it can enhance the infrared light transmittance for the refractive-index matching. The five-layer selective emitter shows very low emissivity at atmospheric windows but strong and broad band absorption at 5–8 μm, perfectly meeting the requirements of infrared camouflage as well as thermal dissipation.

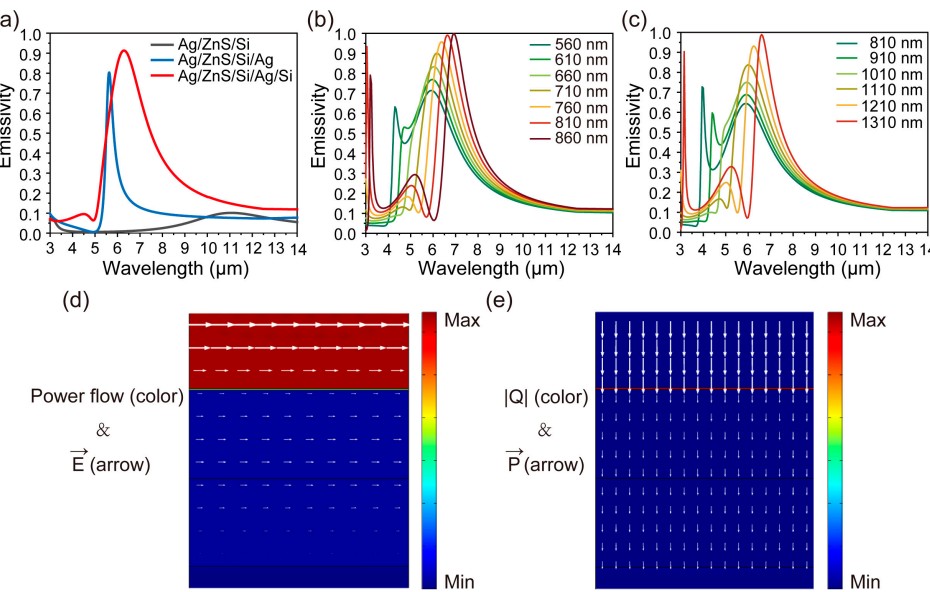

**Figure 3.** (**a**) The spectral absorbance with different structures of Ag/ZnS/Si, Ag/ZnS/Si/Ag, and Ag/ZnS/Si/Ag/Si (100 nm, 410 nm, 410 nm, 350 nm, respectively). The calculated spectral absorbance with different thickness of (**b**) ZnS layer in Ag/ZnS/Ag/Si emitter and (**c**) middle Si layer in Ag/Si/Ag/Si emitter. (**d**) Distribution of the power flow (color) and electric field (arrow) and (**e**) the resistive loss (color) and pointing vector (arrow) at the peak wavelengths of 6.2 μm of the selective emitter. The color maps represent the normalized intensities of power flow and resistive loss, respectively.

The working mechanism of selective thermal emission can be unveiled based on the spectral information, electric field distribution and resistive loss analysis in each layer. As mentioned above, the two Ag layers work as an FP cavity to keeps the incident wave trapped within the composite ZnS/Si dielectric layers. The use of a single dielectric ZnS or Si material is unable to achieve the ideal absorption spectrum. Figure 3b shows the emissivity of Ag/ZnS/Ag/Si four-layer film structure with the variation in intermediate ZnS thickness. When the thickness of ZnS is 710 nm, the emissivity of the four-layer structure at 5–8 and 8–14 um is basically similar to that of the optimal five-layer emitter except that the emissivity at 3–5 μm is relatively higher. The decrease in ZnS thickness below 710 nm results in the appearance of another absorption peak in 3–5 μm. As the thickness exceeds 710 nm, the absorption peak at short wave merges into the main absorption peak accompanying with the redshift and narrowing phenomenon. When the ZnS dielectric layer is replaced by the Si spacer, the Ag/Si/Ag/Si four-layer emitter also exhibits similar emissivity variations such as the ZnS structure (Figure 3c). The spectral difference between the four-layer and optimal five-layer emitter might be attributed to the refractive index matching of the intermediate ZnS and Si dielectric layer.

The electric field, power flow, resistance loss and pointing vector distribution of the Ag/ZnS/Si/Ag/Si five-layer emitter at the resonant peak of 6.2 μm are subsequently analyzed to gain a deeper insight into the physical mechanism underlying the absorption spectra. The energy flow and energy loss density are shown in Figure 3d,e and Figure S2. It can be observed that the electric field intensity is uniformly and strongly distributed in the top Si layer, which is different from other cases using Si as the dielectric layer [25]. When

the thickness of the top Si layer is 450 nm, the incident wave at 6.2 μm can be effectively trapped in the Si layer based on the 1/4 wavelength absorbing principle [26] (Figure S3). The anti-reflective top Si layer not only enhances the transmittance of infrared light to the FP cavity below but also influences the resonance behavior [27]. The function of the ultrathin Ag layer can be investigated from the power flow and resistance loss data. As shown in Figure 3d, the power flow of incident waves rapidly attenuates in the ultrathin Ag layer, and is close to zero in the composite ZnS/Si dielectric layers and the bottom Ag layer. The sparse electric field arrows indicate that the electric field is mainly concentrated in the FP cavity formed by Si/ZnS dielectrics and two metallic Ag layers. In fact, the top Si layer acts as an infrared absorber layer for the ultrathin Ag, allowing more infrared waves to enter the ultra-thin Ag layer inside the emitter, which is consistent with the time-averaged power flow of top Si passing through the ultrathin Ag layer to the bottom FP resonant cavity (Figure 3e). In addition, the distribution of resistive loss density at 6.2 μm also demonstrate that most of the incident wave energy is mainly absorbed in the ultrathin Ag layer. Therefore, the designed selective emitter has higher emissivity around 6.2 μm according to Kirchhoff's law.

Due to the tunneling effect of the ultrathin metal, electromagnetic waves can pass through the ultrathin Ag surface and enter inside of the dielectric layer. By virtue of the FP cavity formed by Ag layers and intermediate Si/ZnS dielectric layers, incident light with specific wavelengths can form stable oscillations, and electromagnetic waves are continuously absorbed by the ultra-thin metal during repeated oscillations, finally realizing the selective radiation of the whole structure. Therefore, FP cavity resonance of partial incident light contributes to the broadband emissivity at the undetected band 5–8 μm. The composite dielectric Si and ZnS layers in the Ag/ZnS/Si/Ag/Si five-layer emitter have very low infrared extinction coefficients, and thus they hardly absorb infrared incident waves. The thickness of the silver film at the bottom is large enough to act as an infrared reflective layer. By comprehensively analyzing the simulation diagram of energy flow and energy loss density, we can find that the ultrathin Ag film near the top is the functional layer for infrared waveband radiation, and the high emissivity characteristics of Ag/ZnS/Si/Ag/Si film in 5–8 μm band mainly come from 7 nm Ag film. Furthermore, the dielectric layers of Si/ZnS control the spectral selectivity by changing the transmission path of electromagnetic waves in the multilayer film.

### 3.2. Experimental Demonstration and Radiative Cooling Performance

3.2.1. The Infrared Emissivity and Radiation Intensity of Selective Thermal Emitter

The Ag/ZnS/Si/Ag/Si five-layer thermal emitter is fabricated by electron beam evaporation on the Si substrate, and the emissivity of sample at different temperatures are measured by a Fourier infrared spectrometer. The emissivity curves and corresponding average emissivity values of the selective emitter are measured in the actual thermal environment with temperatures ranging from 25 °C to 400 °C, and the results are shown in Figure 4a,b. The average emissivity $\varepsilon$ of the sample in the specific wavelength band at temperature $T$ can be expressed by the spectral emissivity and the blackbody radiation as follows:

$$\bar{\varepsilon} = \frac{\int_{\lambda_1}^{\lambda_2} \varepsilon_\lambda E_{b\lambda}(T) d\lambda}{\int_{\lambda_1}^{\lambda_2} E_{b\lambda}(T) d\lambda} \tag{1}$$

$$E_{b\lambda} = \frac{2\pi h c^3}{\lambda^5 exp(hc/k\lambda T - 1)} \tag{2}$$

$$M_{\lambda_1-\lambda_2} = \int_{\lambda_1}^{\lambda_2} \varepsilon_\lambda E_{b\lambda}(T) d\lambda \tag{3}$$

where $E_{b\lambda}$ is the spectral irradiance of the blackbody, $M$ is the integral irradiance, $\lambda_1$ and $\lambda_2$ represent the band range, $\varepsilon_\lambda$ is the emitter emissivity, and $h$, $c$, $k$, $T$ are the Planck's constant, the speed of light in vacuum, the Boltzmann's constant, and temperature, respectively. The

sample effectively maintains its spectral selectivity well all over the measured temperature range (Figure 4a). The measured spectral emissivity in the experiment is basically in consistent with the trend of simulated results as shown in Figure 1b, proving that our emitter design is reasonable. As the temperature increases from 25 °C to 350 °C, the emissivity at 3–14 μm shows a little variation, which indicates that the designed selective emitter has superior stability at low–mid temperatures. Generally, the average emissivity data of the emitters over that whole 3–14 μm remain thermally stable below 350 °C. Specifically, the average emissivity values of the emitter at 3–5 μm and 8–14 μm remain close to 0.15 and 0.2, respectively, when the temperature increases from 150 °C to 350 °C. The average emissivity of the 5–8 μm band is generally stable at 0.7–0.8. With the increase in temperature to 400 °C, the average emissivity of 5–8 μm is basically same as other temperatures, while this value increases sharply to 0.58 at 8–14 μm, which is not in line with expectations of infrared stealth.

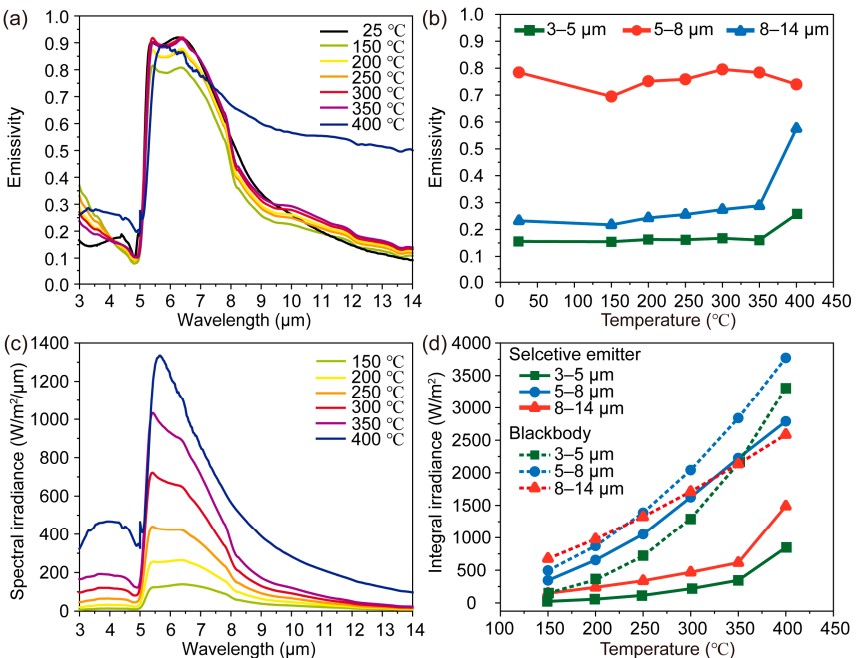

**Figure 4.** (**a**) Measured spectral emissivity at various temperatures ranging from 25 °C to 400 °C; (**b**) The average emissivity in each band (3–5 μm, 5–8 μm and 8–14 μm) of emitters at various temperatures; (**c**) Infrared spectral irradiance of emitters at various temperatures; (**d**) Infrared integrated irradiance of blackbody and emitters at 3–5 μm, 5–8 μm and 8–14 μm at various temperatures, respectively.

Figure 4c,d show the spectral irradiance and integrated irradiance average emissivity of each band calculated by the emissivity measured at different temperatures. The spectral irradiance of the emitter increases sharply as the temperature increases from 150 °C to 400 °C. The increased temperature results in the irradiance peak shifting toward the long-wave direction, and the spectral irradiance band becomes narrow. Furthermore, the integrated irradiance of the selective emitter is investigated in comparison with the black body in the temperature range of 150 °C to 400 °C (Figure 4d). The black body generally shows higher irradiance intensity than the selective emitter over the whole infrared region. By comparing the comprehensive irradiance of the emitter with that of the black body, it is found that the irradiance intensity of the selected emitter in the band of 3–5 μm and 8–14 μm is significantly lower than that of the black body, and the radiation capacity of the selected emitter in the range of 5–8 um is close to that of the black body. In particular, the infrared irradiance reduces by 83.91% for 3–5 μm and 71.18% for 8–14 μm at 350 °C. In addition, the selective emitter shows good radiation heat dissipation at high temperatures, with the spectral irradiance intensity reaching 78.39% and 73.96% of the blackbody value at 350 °C and 400 °C, which guarantees the thermal stability of the designed emitter under

high temperatures. Continuous increase in the ambient temperature over 450 °C will lead to the delamination and destruction of surface films, and the designed five-layer emitter is unable to achieve infrared stealth. The destruction process derives from the emergence of small holes near the ultrathin Ag film, which is supposed to be caused by the Ag oxidation reaction under high temperature. The increased volume of Ag oxides results in expansion and compression to the adjoining layer, and will exacerbate the formation of holes. The continuous expansion of holes finally leads to the delamination of the ultrathin Ag layer. The above experimental results show that the Ag/ZnS/Si/Ag/Si emitter structure can not only achieve a better stealth effect, but is also able to withstand higher temperatures (Figure S4). To sum up, the design of the selective emitter in this work confers a strong heat dissipation capacity and thus ensures good thermal stability and low apparent temperature in the range 25–350 °C, proving the feasibility of selective thermal emission in low/mid-temperature infrared applications.

### 3.2.2. Infrared Camouflage Demonstration of the Selective Thermal Emitter

The infrared camouflage performance of the fabricated emitters is further demonstrated by measuring the radiation temperature of samples in the real thermal environment. Radiative cooling performance is of great significance to infrared stealth, which can be verified through thermal measurements over a range of temperatures. The measurement system is shown as Figure 5a, where an infrared camera is used to monitor the radiation temperature of an object as it rises with the actual temperature. The radiation temperature of the object is controlled by a heating table and a silicon wafer with high emissivity is used as a reference. Figure 5b,c show the thermal images and radiation temperature curves of the Ag/ZnS/Si/Ag/Si five-layer structure and the polished silicon wafer under different heating temperatures from 150 °C and 400 °C. It is clear that the prepared emitter samples show huge reduction in thermal radiation at variable object temperatures compared to the silicon wafers. As the heating temperature gradually increases from 150 °C to 400 °C, the radiative temperatures of the sample and Si substrates rise accordingly. At an object temperature of 200 °C, the radiation temperature of the selective emitter is 115.8 °C, which is much lower than that of the Si wafer under the same conditions (159.4 °C). Notably, along with the increase in heating temperature, the radiation temperature of the sample gradually rises with a much lower growth speed than that of the background heating table, which is attributed to the low emissivity property of the designed emitter in the IR camera's operating band (7.5–14 μm). When the heating temperature reaches to 350 °C, the radiation temperature of sample is only 210.7 °C, corresponding to about 40% reduction in the radiative intensity. The infrared pictures at different temperatures demonstrate that the selective emitter shows low visibility in the infrared camera despite a higher surface temperature, indicating the ideal IR camouflage performance over the wide low/mid-temperature range.

The high emission of the selective emitter in the non-detection band that is responsible for the high heat dissipation and further radiative cooling ability. Figure 5d shows the radiant exitance of the blackbody and emitter sample at 150 °C. The radiation power of the sample is calculated to be 23 W/m$^2$ and 147 W/m$^2$ in the 3–5 μm and 8–14 μm ranges respectively, which is much lower than that of the blackbody. The emitter exhibits good heat dissipation ability with high radiation power of 343.70 W/m$^2$ at band of 5–8 μm, almost 207 times the radiation power of Ag film with very low emissivity. The spectral selectivity of fabricated emitter contributes to the low emission in the detection band and high emission in the non-detection band, which has potential prospects in infrared camouflage as well as thermal management.

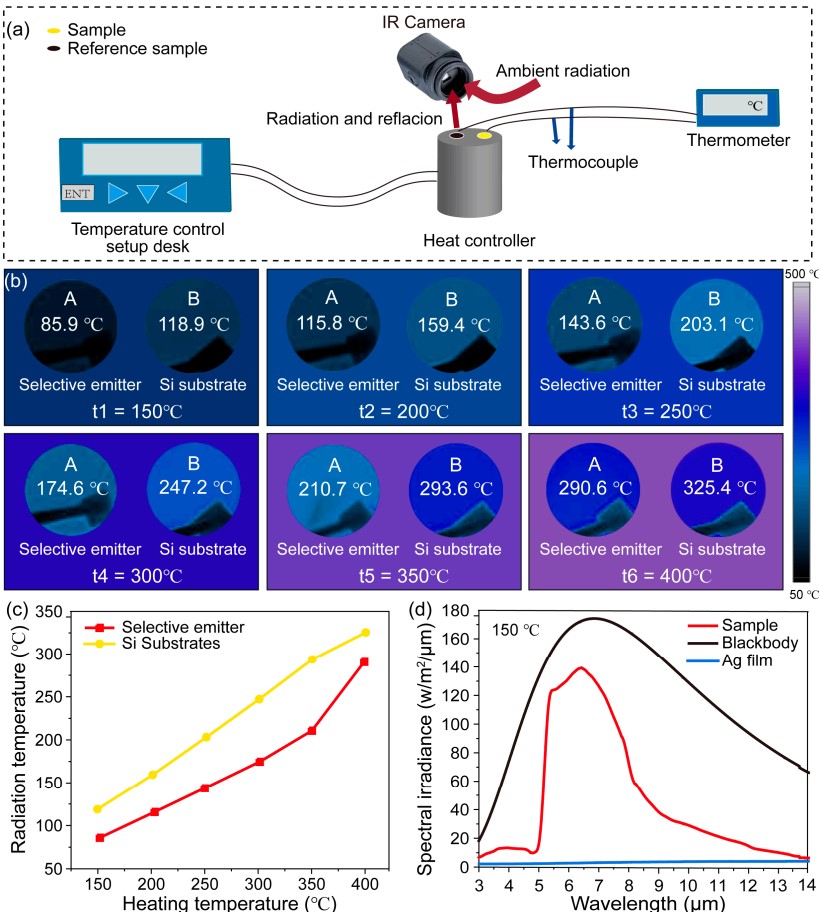

**Figure 5.** (**a**) Schematic of MIR figures detected by the MIR camera; (**b**) Infrared images and radiation temperatures of samples at different background (heating table) temperatures; (**c**) radiation temperatures of samples and high-emissivity objects (silicon substrate) at different heating temperatures; (**d**) Spectral radiance of the sample, reference Ag film and blackbody.

## 4. Conclusions

To summarize, a five-layer film based selective emitter with advantages of simple structure, scalable large-scale fabrication and thermal stability is designed for infrared stealth applications. The emissivity of the fabricated selective emitter is as low as 0.16 and 0.23 in the two atmospheric windows and reaches 0.78 in the 5–8 μm non-atmospheric window at 25 °C. The designed emitter maintains intact structure under high temperatures of 400 °C and the radiative property for infrared camouflage remains stable as the heating temperature is below 350 °C. In comparison to the pure silicon wafer substrate, the emitter shows lower surface radiative temperature thanks to its good dissipation ability. The proposed emitter combines emissivity reduction in the detection band for infrared stealth and emissivity enhancement in the undetected band for heat dissipation. The thermal emitter with a simple multilayer structure and facile fabrication process has good spectral selectivity and thermal stability in the infrared region, which paves the way for the development of selective emitters for radiative cooling and infrared stealth applications.

**Supplementary Materials:** The following supporting information can be downloaded at: https://www.mdpi.com/article/10.3390/photonics10060645/s1, Figure S1: The linear diagram of the simulated emissivities of the emitter under different geomet-ric parameters; Figure S2: Simulation results of the distributions of power loss density; Figure S3: Calculation of the thickness of the anti-reflective silicon layer at the top of the selective emitter based on the 1/4 wavelength absorbing principle; Figure S4: Emissivities and spectral radiance of the reference samples.

**Author Contributions:** Conceptualization, M.Q. and K.Y.; methodology, M.Q.; software, L.Q. and J.H.; validation, K.Y. and Y.L.; formal analysis, M.Q.; investigation, Q.S. and M.Q.; resources, M.Q.; data curation, Q.S. and M.Q.; writing—original draft preparation, M.Q.; writing—review and editing, M.Q.; visualization, C.G.; supervision, K.Y.; project administration, Y.L.; funding acquisition, Y.L. and K.Y. All authors have read and agreed to the published version of the manuscript.

**Funding:** This work was funded by the National Natural Science Foundation of China (Grant Nos. 62105096, 62075058), Science and Technology Project of Henan Province (No. 222102210190), Natural Science Foundation of Henan Province (222300420011), and Program for Innovative Research Team (in Science and Technology) in University of Henan Province (No. 23IRTSTHN013).

**Institutional Review Board Statement:** Not applicable.

**Informed Consent Statement:** Not applicable.

**Data Availability Statement:** The data presented in this study are available on request from the corresponding author.

**Conflicts of Interest:** The authors declare no conflict of interest.

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
