# Peer review of "Fabrication of Selective Thermal Emitter with Multilayer Films for Mid-/Low-Temperature Infrared Stealth with Radiative Cooling"

_photonics, doi:10.3390/photonics10060645_

Round 1
Reviewer 1 Report
The article titled "Fabrication of selective thermal emitter with multilayer films for mid-/low-temperature infrared stealth with radiative cooling" is well-written. The topic is interesting and it provides new information. The conclusions are consistent with the results, and the references are appropriate. The figures clearly illustrate the topic.
However, I have a few minor corrections to suggest to the authors:
- In Figure 3c and d, please add a vertical axis.
- In line 321, please correct the reference to "Figure 5d" instead of "Figure 6c" (as there is no Figure 6c in the article).
- In line 323, I am unable to find the results (22.99 W/m2 and 147.04 W/m2) in Figure 5d. Could the authors provide an explanation for these results in Figure 5d?
Author Response
Comments: The article titled "Fabrication of selective thermal emitter with multilayer films for mid-/low-temperature infrared stealth with radiative cooling" is well-written. The topic is interesting and it provides new information. The conclusions are consistent with the results, and the references are appropriate. The figures clearly illustrate the topic. However, I have a few minor corrections to suggest to the authors:
Response: Thanks a lot for the comments. We have revised our manuscript with care. Changes have been highlighted in the manuscript. All the comments have been carefully addressed and a point-by-point response has been listed as follow.
Comment 1: In Figure 3c and d, please add a vertical axis.
Response: Sincere thanks for your advice. Maybe you refer to the vertical axis of Figure 3d and e. Actually, the power flow distribution and the resistive loss image corresponding to Figure3d and Figure 3d have no vertical axis like other line graphs. And the two images are used to simply demonstrate the power flow/electric field and resistive loss/pointing vector information of selective emitter at resonant wavelength. And the intensity of power flow and resistive loss are normalized and shown in the color map, which is supplemented in the caption of Figure 3.
(Line 207) The color maps represent the normalized intensities of power flow and resistive loss, respectively.
Comment 2: In line 321, please correct the reference to "Figure 5d" instead of "Figure 6c" (as there is no Figure 6c in the article).
Response: Sincere thanks for your kind advice. The reference content has been revised as “Figure 5d”.
Comment 3: In line 323, I am unable to find the results (22.99 W/m2 and 147.04 W/m2) in Figure 5d. Could the authors provide an explanation for these results in Figure 5d?
Response: Thank you for your question. The irradiation power of 22.99 W/m2 and 147.04 W/m2 cannot be directly observed in Figure 5d. They are integrals of spectral irradiance in the bands of 3-5 um and 8-14 um shown in Figure 5d, respectively.
Reviewer 2 Report
The article “Fabrication of selective thermal emitter with multilayer films for mid-/low-temperature infrared stealth with radiative cooling” present the theoretical and experimental investigations of an Ag/ZnS/Si/Ag/Si multilayered emitter. Proposed emitter has a low emissivity in IR bands (3-5 µm and 8-14 µm), high emissivity at 5-8 µm and maintains excellent stealth performance until 350°C. These properties make him a good candidate for the facile fabrication of mid-/low-temperature infrared stealth devices.
My questions and suggestions:
Line 123 numerical analysis subsequently performed to further investigate the effects of geometrical parameters on spectral behaviors.
Line 130-131 the simulated results are shown in Figure 2 and Figure S1.
Please describe how you made the numerical simulations of absorption spectra and how you calculated spectral absorbance? It is worth to write a one paragraph about it how this calculation was done.
This work is probably worthy of publication in Photonics (publ. MDPI) with major corrections listed as a comments/questions in attached pdf files and placed on the MDPI server in Report Review Form.

All linguistic errors that I found in the text of the manuscript were marked with special notes and comments in the pdf file of the manuscript (photonics-2414166-peer-review-v1.pdf).
Author Response
Comments: The article “Fabrication of selective thermal emitter with multilayer films for mid-/low-temperature infrared stealth with radiative cooling” present the theoretical and experimental investigations of an Ag/ZnS/Si/Ag/Si multilayered emitter. Proposed emitter has a low emissivity in IR bands (3-5 µm and 8-14 µm), high emissivity at 5-8 µm and maintains excellent stealth performance until 350°C. These properties make him a good candidate for the facile fabrication of mid-/low-temperature infrared stealth devices.
Response: Thanks a lot for the comments. We have revised our manuscript with care. Changes have been highlighted in the manuscript. All the comments have been carefully addressed and a point-by-point response has been listed as follow.
Comment 1: Line 123 numerical analysis subsequently performed to further investigate the effects of geometrical parameters on spectral behaviors.
Response: Sincere thanks for your advice. This sentence has been revised.
Comment 2: Line 130-131 the simulated results are shown in Figure 2 and Figure S1.
Response: Sincere thanks for your kind advice. This sentence has been revised.
Comment 3: Please describe how you made the numerical simulations of absorption spectra and how you calculated spectral absorbance? It is worth to write a one paragraph about it how this calculation was done.
Response: Thank you for your question. The calculation process has been supplemented in the revised manuscript in the Materials and Methods part (as follow).
(Line 74) Numerical simulations: the optical simulation was performed with a wave optical module (electromagnetic waves, frequency domain) in COMSOL Multiphysics 6.0 software. The governing equation was Maxwell’s equation and it was discretized for a finite element method. The simulated domain consisted of a metal ground (Ag), a dielectric layer (ZnS), a lower dielectric layer (Si), a metal layer (Ag), an upper dielectric layer (Si) and a medium (Air). The parameters of the structure were defined in the text. We adopted the Lorentz-Drude model for analyzing the electromagnetic behaviors of Ag. The refractive index in COMSOL Multiphysics 6.0 software of ZnS and Si were used to model for dielectric layers. The real and imaginary parts of the refractive index of medium were assumed to be 1 and 0, respectively. Planar electromagnetic waves at the two ports were assumed, using periodic boundary conditions for the side walls. The emissivity (E) of the multilayer films is determined by calculating the absorptivity (A) according to the Kirchhoff’s law which states that the spectral absorptivity and emissivity must be equal to each other (E = A).
Reviewer 3 Report
In the manuscript titled "Fabrication of Selective Thermal Emitters with Multilayer Films for Mid-/Low-Temperature Infrared Stealth and Radiative Cooling," the authors present a multilayer selective thermal emitter with low emissivity in the 3-5 µm and 8-14 µm bands, and high emissivity in the 5-8 µm band. This unique spectral emissivity profile makes the emitter well-suited for applications in infrared camouflage and radiative cooling. The design principles behind the emitter are thoroughly discussed, and the authors provide corresponding numerical simulations. Experimental characterization of the emitter's emissivity reveals lower radiative temperatures compared to a reflective sample in the two desired IR bands. Overall, the work is compelling, and the experiments are robust. I recommend publication upon addressing the following concerns:
1) The authors argue that their multilayer structure is more practical than other emitters featuring micro/nano-structures. How would they compare their work with other multilayer structures, particularly the widely-used ZnS/Ge multilayers? For reference, consider these publications: [Light: Science & Applications, volume 9, Article number: 60 (2020)] and [Nature Communications, volume 12, Article number: 1805 (2021)].
2) The potential for infrared camouflage of the emitter is demonstrated by comparing its radiative temperature to that of a reference sample. The authors link this result to radiative cooling. However, a more persuasive argument for radiative cooling would involve quantifying the absolute temperature of the sample and the reference. Can the authors provide information, either experimental or theoretical, on how much cooler the emitter would be compared to the reference, given its dissipation channel in the 5-8 µm band?
3) The 400°C temperature limit is mentioned, but the fundamental reasons for this constraint should be discussed. Beyond stating "delamination and destruction," the authors should provide more detailed explanations, such as the material failure mechanisms. Based on this discussion, how can the reliability of the emitter be improved in the future?
Round 2
Reviewer 2 Report
There is one typos at line no 88: word "emtter", instead of "emitter".
I accept the article in the new, corrected version.
No comments.
Author Response
The minor mistake has been revised and the manuscript is updated.